# OUTPUT-CONSTRAINED DECISION TREES

## ABSTRACT

When there is a correlation between any pair of targets, one needs a prediction method that can handle vector-valued output. In this setting, multi-target learning is particularly important as it is widely used in various applications. This paper introduces new variants of decision trees that can handle not only multi-target output but also the constraints among the targets. We focus on the customization of conventional decision trees by adjusting the splitting criteria to handle the constraints and obtain feasible predictions. We present both an optimization-based exact approach and several heuristics, complete with a discussion on their respective advantages and disadvantages. To support our findings, we conduct a computational study to demonstrate and compare the results of the proposed approaches.

## 1 INTRODUCTION

Decision trees (DTs) play a fundamental role in machine learning and data science for several key reasons. First, DTs have proven to be accurate predictive models in a wide range of applications. Moreover, the heuristic approaches used for their construction are computationally efficient, resulting in extremely fast training times. Accuracy and speed make them ideal for large datasets or real-time applications. Second, DTs, particularly those that are not too deep, are inherently explainable. By following the branches of such a trained tree, one can easily understand the logic behind the model's predictions. This feature is particularly valuable in sectors where interpretability is critical, such as in healthcare or finance. Lastly, DTs serve as the base learners in powerful ensemble methods. These methods, which include techniques like random forests and gradient boosting, utilize numerous DTs to create robust and accurate predictive models.

When there is a correlation between any pair of targets, we need a prediction method that can handle vector-valued output. In this setting, multi-target learning is particularly important as it is widely used in fields such as forecasting, monitoring, manufacturing, and recommendation systems. DTs can easily be customized for handling multi-target regression by modifying the splitting criteria to handle vector predictions at each node.

**Motivation.** Despite the wealth of existing literature on DTs, to the best of our knowledge, none consider cases where constraints exist among multiple targets. This is notable, as in applications, decision-makers are aware of these constraints, and data analysts should impose them for *feasible* predictions. Here is an illustrative example from the field of operations management where machine learning approaches are becoming more prominent in the context of decision-making; see *e.g.*, (Mišić & Perakis, 2019).

Consider a dataset of products in a retail store, each with distinct attributes. This dataset includes the inventory amounts of these products across five different warehouses, represented as five-dimensional target vectors (training set). The goal is to predict the inventory amounts for new products across these warehouses (test set).

To achieve this, one can train a DT and obtain the partitions of the existing products represented by the leaf nodes of the trained tree. The prediction for a new product is then simply the average of the inventory amounts (mean vector) within the leaf node that it falls into. This leads to the prediction of five different inventory amounts, *i.e.*, one vector for each warehouse.

However, it is crucial to consider an additional constraint in our scenario: Due to budget restrictions, a store can only transport a product in a maximum of two out of the five warehouses. This is also evident in the training data set where each target vector has at most two nonzero values. Consequently,

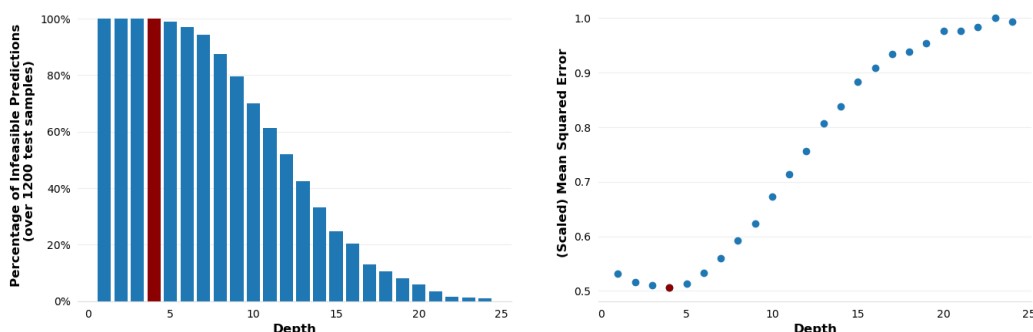

Figure 1: The trade-off between accuracy and feasibility. The red circle in the right plot marks the depth of the tree which returns the lowest mean squared error value for the test set. The corresponding red bar in the left plot shows the corresponding percentage of infeasible predictions. The sample size is 4000.

if one continues to use the mean vector approach for prediction, it may not only yield inaccurate results but also infeasible predictions because it does not account for this constraint.

This erroneous estimation would also persist with many well-known DT training algorithms, since their estimation at a leaf node is also based on taking the mean or the median of the target vectors within the node. However, one may argue that growing a deeper DT may resolve the infeasibility issue as there will be fewer products at each leaf node. Even if we overlook the loss of interpretability due to the large size of such a tree, it is well-known that deep trees can easily overfit to the training data, potentially leading to poor generalization for new products. For instance, Figure 1 shows our results with one of our datasets from our computational study in Section 3. The bar plot demonstrates that feasible predictions can be achieved with larger tree depths, but this comes with a significant decrease in accuracy as shown in the scatter plot. The scatter plot, on the other hand, shows that while shallow trees offer much better accuracy at the expense of most of their predictions are being infeasible.

**Output-Constraints.** To obtain accurate and feasible predictions, we propose an optimization-based exact approach. This approach incorporates target constraints into the splitting procedure of common binary tree construction methods. Our approach ensures that the estimated target vector at each node meets the required constraints. However, we observe that the exact approach could have a lengthy computation time, especially for deep trees. To address this, we offer two heuristic approaches. These are based on using existing samples as predictions or relaxing the constraints with a penalty function at each split operation.

Up until now, our focus has been on regression problems. It is reasonable to question why we have not considered multi-target classification problems as well. The reason is, if the number of classes for each target and the number of targets are relatively small, we can assign a label to each *feasible* combination of target values. This approach effectively reduces the problem to a standard single-target classification problem, although possibly with many labels.

**Related Work.** As machine learning methods improve in accuracy, they may still produce predictions and outcomes that lack physical consistency or feasibility, due to biases or data sparsity (Zhang et al., 2022). To overcome this challenge, hybrid methods that integrate optimization techniques with machine learning models have gained significant attention in recent years due to their ability to leverage the strengths of both approaches. One key aspect of hybrid approaches is their ability to handle constraints from various sources to enhance the feasibility and accuracy of machine learning methods. These constraints can originate from the dataset, user-defined specifications, or domain knowledge (Nanfack et al., 2022). Such machine learning approaches with imposed constraints are developed as practical applications across different domains, such as mechanics (Yan et al., 2022), geomechanics (Chen & Zhang, 2020), and electronics (Lee et al., 2019).

Traditional supervised learning methods primarily focus on predicting a single output, using data comprised of instances represented by features and a single response variable. Multi-target regression (also referred to as multi-output regression), is used to predict multiple continuous target variables simultaneously. One of the earliest methods for multi-target regression with decision trees was introduced by De'ath (2001). He extended the univariate recursive partitioning approach from Classification and Regression Trees (CART) (Breiman, 2017) to handle multiple responses, calling this method Multi-Response Trees (MRT). MRTs are built similarly to CART, but the node impurity is redefined as the sum of squared errors across all response variables. Struyf & Džeroski (2006) developed a system for building multi-objective regression trees (MORTs), which include constraints on tree size and accuracy, allowing users to balance interpretability and accuracy by limiting tree size or setting a minimum accuracy threshold. (Adıyeke & Baydoğan, 2022) introduced a novel method combining multi-target regression with unsupervised learning. Specifically, they modified tree construction by incorporating a similarity-based clustering indicator in ensemble methods.

Decision trees (DTs) offer some flexibility in enforcing constraints, and many hybrid DT approaches incorporate *instance-level* constraints to bind a subset of instances in generating meaningfully combined results. For instance, the clustering tree algorithm by Struyf & Džeroski (2007) integrates domain knowledge to impose must-link and cannot-link constraints. DT methods have also been developed to address adversarial examples, which can be considered instance-level constraints (Chen et al., 2019; Calzavara et al., 2019), or attribute-level constraints to assign several features to the predictors (Nanfack et al., 2022). Ben-David (1995) was one of the first to apply *attribute-level* constraints by imposing monotonicity among the model's predictors, and Cardoso & Sousa (2010) extended this in an ordinal classification setting.

One of the earliest hybrid approaches of imposed constraints was implemented by Lee & Kang (1990), who introduced constraints to deep learning architectures to approximate ordinary and partial differential equations. This approach laid the foundation for the general framework known as physics-informed neural networks, proposed by Raissi et al. (2019), where constraints and boundary conditions are incorporated (Rudin et al., 2022). Similarly, Christopher et al. (2024) integrated constraints and physical principles to generative diffusion processes. To the best of our knowledge, our proposed approach is the first method designed to handle *target-level* constraints in multi-target regression. This represents a novel contribution, building on the principles of constraint handling and extending them to more complex, interdependent target structures in predictive modeling.

**Contributions.** We introduce novel approaches to handle constraints among target variables in DTs. The first method lends itself to an optimization-based exact method that ensures the feasibility of predictions by incorporating target constraints into the splitting procedure of common binary tree construction methods. We also propose two heuristic methods. These methods offer practical solutions to the computation time issue that might be associated with the exact approach, especially for deep trees. We provide the technical details of our proposed methods, exploring their strengths, potential limitations, and the conditions under which they would perform sensibly. To validate our approaches, we provide a reproducible computational study using our publicly available implementation.

## 2 OUTPUT-CONSTRAINED DECISION TREES

Suppose that we have the training dataset $\mathcal{D} = \{(\boldsymbol{x}_i, \boldsymbol{y}_i) : i \in \mathcal{I}_{\mathcal{D}}\}$ with $\boldsymbol{x}_i \in \mathbb{R}^p$ and $\boldsymbol{y}_i \in \mathbb{Y} \subseteq \mathbb{R}^k$ denoting the input vector and the target (output) vector for the data point $i$, respectively. The set $\mathbb{Y}$ shows the *feasible set* for the target vectors. It is important to observe that the target vector of each sample lies in $\mathbb{Y}$, and hence, they are also feasible.

A DT is grown by splitting the dataset into non-overlapping subsets recursively (Breiman et al., 1984). Each subset is represented by a node of the tree. The most common approach of tree construction uses binary splits; that is, at each iteration at most two child nodes, left ($\leftarrow$) and right ($\rightarrow$), are grown from a parent node. Let $\mathcal{N} \subseteq \mathcal{D}$ be the subset denoted by the parent node, and $\mathcal{I}_{\mathcal{N}} \subseteq \mathcal{I}_{\mathcal{D}}$ indicate the indices of the samples in that node. To obtain a pair of child nodes, a feature $j$ and a split value $v$ is selected to obtain the left and right index sets

$$\mathcal{I}_{\mathcal{N}}^{\leftarrow}(j, v) = \{i \in \mathcal{I}_{\mathcal{N}} : x_{ij} \leq v\}$$

and

$$\mathcal{I}_{\mathcal{N}}^{\rightarrow}(j, v) = \{i \in \mathcal{I}_{\mathcal{N}} : x_{ij} > v\},$$

respectively. Then, the gain by such a split is evaluated by

$$\alpha_{\mathcal{N}}^{\leftarrow}(j,v) \sum_{i \in \mathcal{I}_{\mathcal{N}}^{\leftarrow}(j,v)} \ell(\boldsymbol{y}_{\mathcal{N}}, \boldsymbol{y}_i) + \alpha_{\mathcal{N}}^{\rightarrow}(j,v) \sum_{i \in \mathcal{I}_{\mathcal{N}}^{\rightarrow}(j,v)} \ell(\boldsymbol{y}_{\mathcal{N}}, \boldsymbol{y}_i),\qquad(1)$$

where $\boldsymbol{y}_{\mathcal{N}} \in \mathbb{Y}$ is the prediction of the parent node, $\ell : \mathbb{R}^k \times \mathbb{R}^k \mapsto \mathbb{R}$ is the loss function, $\alpha_{\mathcal{N}}^{\leftarrow}(j,v) = \frac{|\mathcal{I}_{\mathcal{N}}^{\leftarrow}(j,v)|}{|\mathcal{I}_{\mathcal{N}}|}$ as well as $\alpha_{\mathcal{N}}^{\rightarrow}(j,v) = 1 - \alpha_{\mathcal{N}}^{\leftarrow}(j,v)$ are the scaling coefficients representing the proportional weights of the respective subsets. Among all possible splits, the one that achieves the largest gain is used to construct the child nodes. The splitting continues until one of the stopping conditions, such as reaching the maximum depth or the minimum number of samples in a node, is satisfied. Then, the samples in the last parent constitute a leaf node. Considering the feasibility requirements on the predictions, the output vector $\boldsymbol{y}_{\mathcal{N}}$ of such a leaf node should be feasible; *i.e.*, it should belong to the feasible set, $\boldsymbol{y}_{\mathcal{N}} \in \mathbb{Y}$.

## 2.1 EXACT APPROACH

The common loss functions (*e.g.*, mean squared error, mean Poisson deviance) used in constructing binary DT regressors are separable. Moreover, the loss in the multi-target case is obtained by summing up individual losses for each target. Consider, for instance, the well-known mean squared loss for which the prediction at a node with the subset $\mathcal{N}$ is given by

$$\boldsymbol{y}_{\mathcal{N}} = \arg\min_{\boldsymbol{y} \in \mathbb{Y}} \frac{1}{2|\mathcal{I}_{\mathcal{N}}|} \sum_{i \in \mathcal{I}_{\mathcal{N}}} \|\boldsymbol{y} - \boldsymbol{y}_i\|^2.$$

If there are no restrictions on the output vectors, *i.e.*, $\mathbb{Y} = \mathbb{R}^k$, then it is not difficult to observe by simple differentiation that $\boldsymbol{y}_{\mathcal{N}}$ is the mean of the target vectors in $\mathcal{N}$ minimizing the subset variance. This is, in fact, the output of a node used by the well-known implementations, like `scikit-learn` (Pedregosa et al., 2011). The same holds for the mean Poisson deviance loss function. In fact, the following theorem shows that the mean vector is *still* the optimal choice when the constraints on the output vectors result in a convex feasible set. The proof of this theorem follows from observing that the mean operation is equivalent to taking the convex combination of the feasible target vectors in the node. The details of this proof are given formally in Section A of the supplementary file. There, we also present a counter example showing that the same argument does not apply when the loss function is the mean absolute deviation and its unconstrained minimum, *i.e.*, the median of the target vectors, is selected as the node's output (see Remark 1 in Section A of the supplementary file).

**THEOREM 1** *Let the feasible set $\mathbb{Y} \subseteq \mathbb{R}^k$ be convex. If the loss function is the mean squared error or the mean Poisson deviance, then the mean of the target vectors in a node minimizes the loss.*

This theorem implies that a restricted class of feasible sets can be handled with standard DT implementations using certain loss functions. For instance, simple bounds or, more generally, linear constraints on the target vectors lead to convex feasible sets. However, the same conclusion does not hold when the feasible set is not convex. Going back to our illustrative example of inventory amount prediction, suppose for simplicity that each warehouse has the same capacity of $C$ units. If we use only these bounds as constraints, then any standard implementation returning the mean vector for each node output would not violate this constraint. However, if we also impose the budget restriction "at most two among five warehouses can be used for transportation," then we need auxiliary binary variables to represent this restriction as a constraint, and hence, the feasible set $\mathbb{Y}$ becomes non-convex. Formally,

$$\mathbb{Y} = \left\{ \boldsymbol{y} \in \mathbb{R}^5 : \sum_{t=1}^{5} z_t \le 2,\ 0 \le y_t \le C z_t,\ \text{and}\ z_t \in \{0,1\}\ \text{for}\ t = 1, \ldots, 5 \right\},\qquad(2)$$

where the binary variable $z_t$ becomes one if $y_t$ units of the product is transported to warehouse $t$. Note that when $z_t$ is zero, then it also forces $y_t$ to become zero implying that product is not transported to the corresponding warehouse.

We can now state the formal prediction model that needs to be solved at each node:

$$\boldsymbol{y}_{\mathcal{N}} = \arg\min_{\boldsymbol{y} \in \mathbb{Y}} \sum_{i \in \mathcal{I}_{\mathcal{N}}} \ell(\boldsymbol{y}, \boldsymbol{y}_i),\qquad(3)$$

where $\mathcal{I}_\mathcal{N} \subseteq \mathcal{I}_\mathcal{D}$ with $\mathcal{N} \subseteq \mathcal{D}$. Dealing with nonconvex sets can pose a challenge when solving large-scale constrained models. Fortunately, the dimension of the problem ($k$) is quite small for many applications. In case of a mixed integer linear optimization problem, like our example on warehouse inventories, both free and commercial powerful solvers are available, *e.g.*, COIN-OR (Lougee-Heimer, 2003), GUROBI (Gurobi Optimization, LLC, 2023). However, it is also important to remember that this optimization problem must be solved for each pair of feature-split values at every node. This increases the computation time compared to basic DT implementation that ignores the constraints, especially for deep trees. Next, we will also propose several heuristic approaches to develop faster –albeit potentially less accurate or infeasible– alternatives.

## 2.2 Heuristic Approaches

A common method for training machine learning models with restrictions involves relaxing these constraints and adding a term to the loss function. This term serves to penalize predictions that are not feasible. In our setting, this idea leads to the following model:

$$\boldsymbol{y}_\mathcal{N}^R = \arg \min_{\boldsymbol{y} \in \mathbb{R}^k} \sum_{i \in \mathcal{I}_\mathcal{N}} \ell(\boldsymbol{y}, \boldsymbol{y}_i) + \lambda v_\mathbb{Y}(\boldsymbol{y}), \tag{4}$$

where $\lambda > 0$ is the predefined *penalty coefficient*, and $v_\mathbb{Y} : \mathbb{R}^k \mapsto \mathbb{R}_+$ is the penalty function that assigns a positive value to the infeasible predictions, $\boldsymbol{y} \notin \mathbb{Y}$. This technique is a reminiscent of the approach known as Lagrangean relaxation. However, it is important to note that Lagrangean relaxation would introduce a separate penalty coefficient, known as Lagrangean multiplier, for each constraint (Bertsekas, 1982). The objective function of (4) shows the trade-off between accuracy and feasibility. For instance, using the feasible region defined in (2) leads to the following splitting problem for the mean-squared loss:

$$\boldsymbol{y}_\mathcal{N}^R = \arg \min_{\boldsymbol{y} \in \mathbb{R}^5, \boldsymbol{z} \in \{0,1\}^5} \frac{1}{2|\mathcal{I}_\mathcal{N}|} \sum_{i \in \mathcal{I}_\mathcal{N}} \|\boldsymbol{y} - \boldsymbol{y}_i\|^2 + \lambda \sum_{t=1}^5 \left( (z_t - 2) + (y_t - Cz_t) \right). \tag{5}$$

Here, $\lambda$ in (4) plays the role of balancing optimality versus feasibility. As $\lambda$ increases, the optimization algorithm tends to favor feasible solutions over accurate ones. Conversely, for small values of $\lambda$, accuracy is prioritized. Therefore, one is not guaranteed to obtain a feasible solution with the relaxation approach. Additionally, depending on the structure of the penalty function $v_\mathbb{Y}$, problem (4) can be difficult to optimize. Note that tuning $\lambda$ requires attention as the optimization problem changes at each node. Attempting to adjust a different penalty coefficient for each node could quickly become a computationally formidable task.

Our second heuristic is based on observing that all samples in a node belong to the feasible set. Thus, selecting one of them results in a feasible prediction. Considering the spread of the output values in the subset, a natural candidate could be the *medoid* as it would be a representative sample within the cluster designated by the node. The medoid of the subset is obtained by solving

$$\boldsymbol{y}_\mathcal{N}^M = \arg \min_{\boldsymbol{y}_{i'} \in \mathcal{N}} \sum_{i \in \mathcal{I}_\mathcal{N}} \ell(\boldsymbol{y}_{i'}, \boldsymbol{y}_i). \tag{6}$$

The solution to this problem involves evaluating the pairwise loss values and then summing these values for each sample. As a result, the medoid of a subset of samples can be found very quickly. In case there are multiple samples that could be the medoids, one of them can be selected arbitrarily. Clearly, the medoid at each node is a feasible prediction but not necessarily the optimal solution of (3). At this point, we note that a tree can be grown very deep so that each leaf node contains a single sample. While this allows for feasible predictions, such deep trees are susceptible to overfitting and can quickly lose interpretability (see also Figure 1).

Similar to the exact approach of the previous subsection, both heuristic approaches above are applied to every feature and split value at each node. An alternative heuristic would be growing a tree as in basic implementation without considering the constraints. Then, the feasibility of DT's prediction can be guaranteed by solving the exact model (3), the medoid model (6), or the relaxation model (4) –with an appropriate choice of the penalty coefficient– at the *leaf nodes only*. This approach would improve the training time significantly.

## 3 COMPUTATIONAL STUDY

This section is dedicated to testing the performance of our exact and heuristic approaches proposed for training output-constrained decision trees (OCDTs). We obtain different implementation variants by using one of the proposed approaches at a decision node for *splitting* (S) or at a *leaf* (L) node for prediction. We also use the shorthand notation O, R, M, A for the exact *optimization*, *relaxation*, *medoid*, and *average*[1] approaches, respectively. For instance, the variant that uses relaxation approach for splitting and the medoid approach for prediction is denoted by `S(R):L(M)`. Likewise, `S(A):L(A)` simply refers to standard DT implementation in software packages that ignores the constraints. In our study, we have used the `scikit-learn` implementation *only for this variant* (Pedregosa et al., 2011).

Our objective in this section is two-fold: First, we aim to assess the effectiveness of our proposed approaches on an existing dataset. Second, we aim to evaluate their performance in a controlled setting using an extended version of the same dataset. Our computational experiments can be reproduced by using the dedicated GitHub repo [2]. For all methods compared, we experimented with varying maximum depth values of 3, 6, 9, 12, and 15, while ensuring a minimum of 10 data points to split a node and at least 5 data points in each leaf. The selection of penalty coefficients, $\lambda$, for each variant can be guided by evaluating several values over a reasonable range through cross-validation. Ideally, the chosen coefficient should minimize MSE while ensuring that no infeasible predictions are generated. The results were obtained using a computer equipped with an Intel(R) Core(TM) i7-9750H CPU 2.60GHz processor, featuring 12 CPUs, and 64GB of RAM.

### 3.1 SCORE DATASET

As a part of our computational study, we employ the *scores* dataset available online (Kimmons, 2012). This dataset consists of 1000 records, simulating various personal and socio-economic factors of students at a public school. The dataset features three target variables, representing math, reading, and writing scores. To tailor the data to our research objectives, we have imposed two constraints: First, if the reading score falls below 50, the writing score is set to zero. Second, if the combined reading and writing scores amount to less than 110, the math score is also set to zero. The full description of our mathematical model is available in Section B of the supplementary file.

We conducted a thorough five-fold cross-validation experiment on the dataset using five different tree-depth parameter values, employing a total of nine distinct variants which are trained with the same parameters. These variants are classified into two primary components: split evaluation and leaf predictions. Split evaluation entails the assessment of each candidate for binary splitting based on the prediction generated by a specific method. Conversely, leaf prediction refers to the prediction method employed by each leaf node. For instance, the methodology involving the medoid split evaluation and optimal leaf prediction entails that each split during tree growth is determined by evaluating the total mean squared error (MSE) values of the resulting child nodes using their medoids as predictions. Subsequently, the resulting DT solves an optimization problem specific to the dataset for data falling to each leaf to assign the prediction of that particular node. The penalty coefficient $\lambda$ is selected by monitoring the trade-off between feasibility and accuracy through five-fold cross-validation.

We evaluated the performance of each variant based on the MSE value obtained during the cross-validation process. Our analysis focused on comparing the performance of an OCDT variant with different split evaluation and leaf prediction approaches against the baseline DT implementation (`scikit-learn`). The results are visualized in the left side of Figure 2 illustrating the key metrics concerning the performance of the variants. The bar plot reflects the (scaled) average MSE values with the left axis indicating the corresponding values. On top of each bar, the MSE values are displayed alongside the average number of infeasibilities resulting from the methods' predictions (in parenthesis). The scatter plot shows the average training time of each method with the right axis denoting the corresponding values.

In general, it appears that it is important to assess the performance of each variant individually. We find that the variant utilizing relaxation in the optimization problem in both split evaluation and leaf prediction, `S(R):L(R)` exhibits the worst MSE score. However, the variant employing relaxation

---

[1]This refers to the standard implementation of taking average of the target vectors of the samples in a node.
[2]https://anonymous.4open.science/r/ocdt-F2EE

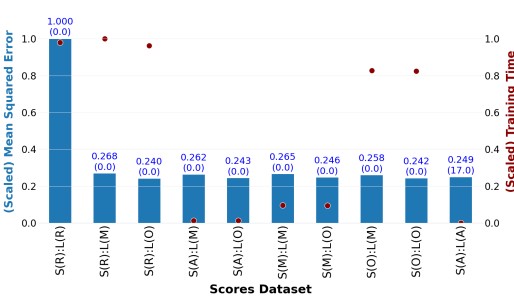 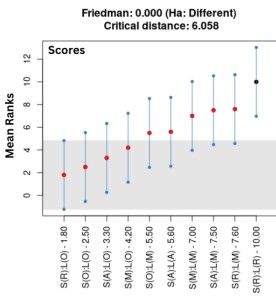

Figure 2: Performance metrics plot (left) and the Friedman test for statistical differences (right) on the *scores* dataset. The values on each bar represent the average MSE value and the average number of infeasible predictions (in parenthesis) on the test sets. The scatter plot shows the training time for each variant. Note that MSE and computation times are scaled according to the maximum value for the corresponding performance measure obtained by one of the variants.

in split evaluation and optimization in leaf prediction, S(R):L(O) achieves the best MSE score without violating any constraints. Overall, we observe that the variants solving optimization problems at the leaf nodes for predictions demonstrate the top three performances for this dataset with no constraint violations. This affirms both the efficiency of the proposed variants and their adherence to constraints. We note that the baseline scikit-learn implementation, S(A):L(A) violated numerous constraints (averaging 16.4 infeasible predictions), aligning with our earlier findings.

The comparison of the proposed variants for statistical significance is conducted using the Friedman test (Friedman, 1940), and if a significant difference is detected by the Friedman test, it is followed by the Nemenyi test (Nemenyi, 1963). The right side of Figure 2 shows the test results. We observe that the *p*-value is practically zero indicating that the null hypothesis of identical variants can be rejected at the 0.95 confidence level. This means that there is a statistically significant difference in the performances of the variants with S(R):L(O) having a notably better performance when compared against the others.

## 3.2 EXTENDED DATASETS

In this section, we extended the previous dataset (Kimmons, 2012) to mimic the motivating inventory control example that we have given in Section 1. To test the proposed approaches in this controlled environment, we imposed other restrictions on the dataset, like each target vector having at most one nonzero value. The details of our data generation process along with the underlying mathematical model is given in Section C of the supplementary file. We call this dataset *class*.

In accordance with the experiments in Section 3.1, we apply the same methodology with five-fold cross-validation. This is done to compare our nine variants using the same parameters. To leverage the advantage of data generation, we evaluated our proposed variants across nine different datasets, varying in size ($|\mathcal{I}_\mathcal{D}|$) and the number of targets ($k$). These datasets comprised of combinations of 500, 1000, and 2000 samples with total number of target values of 3, 5 and 7.

The MSE score and computation time are depicted on the left side of Figure 3, where the average metrics across all nine datasets are given. On the right side of Figure 3, we see that *p*-value is close to 0, which indicates that the methods have statistically different performances at the 0.95 confidence level. We observe that none of the proposed algorithms produce infeasible predictions, unlike the baseline scikit-learn method, S(A):L(A) which, despite yielding the lowest MSE among all approaches, generated the only positive average number of infeasible predictions. This suggests that the scikit-learn method compromised constraints to improve the prediction performance, a trade-off that is clearly not suitable for applications.

The box plot in Figure 4 illustrates the MSE values of all the variants across nine datasets with different sizes in terms of sample-sizes and targets. The proposed variants show comparable MSE scores, with most of the methods achieving scores within a close range, as indicated by the overlapping

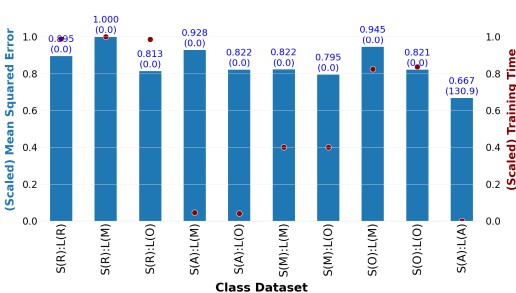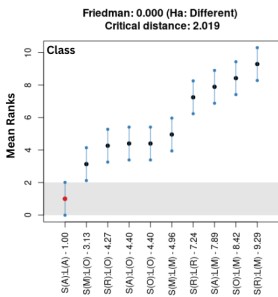

Figure 3: Performance metrics plot (left) and the Friedman test for statistical differences (right) on *class* datasets. The values on each bar of the left plot represent the average MSE value and the average number of infeasible predictions (in paranthesis) on the test sets. The scatter plot shows the training time for each variant. Note that MSE and computation times are scaled according to the maximum value for the corresponding performance measure obtained by one of the variants.

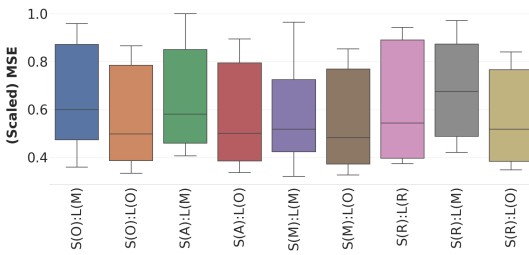

Figure 4: MSE box plot on the *class* datasets.

interquartile ranges. While there is some variation, the results suggest that none of the methods stand out as consistently superior in terms of accuracy.

Figure 5 depicts that the training times of the variants vary significantly with those involving optimization and relaxation, such as S(R):L(R) and S(R):L(M), taking notably longer. This observation aligns with the computational complexity expected from optimization-based approaches. Another noteworthy observation is that the methods with relatively short training times, like S(A):L(M), tend to have higher MSE values. This trade-off between model complexity and training time versus prediction accuracy is evident across the board. Furthermore, variants like S(O):L(M) strike a balance, achieving moderate MSE values while maintaining competitive training times, suggesting a promising option for practical use in time-sensitive applications.

Figure 6 illustrates a schematic representation of a tree trained on a *class* dataset. This tree structure is obtained by solving the optimization problem for the split evaluation as well as for the leaf prediction; *i.e.*, the selected variant is S(O):L(O). In this figure, the green nodes signify the leaf nodes, while the red nodes represent the decision nodes where the data is split. Additionally, the splits after each decision node are depicted with arrows indicating the left and the right child nodes, accompanied by

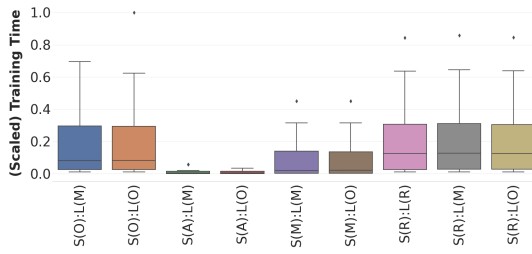

Figure 5: Training time box plot on the *class* datasets.

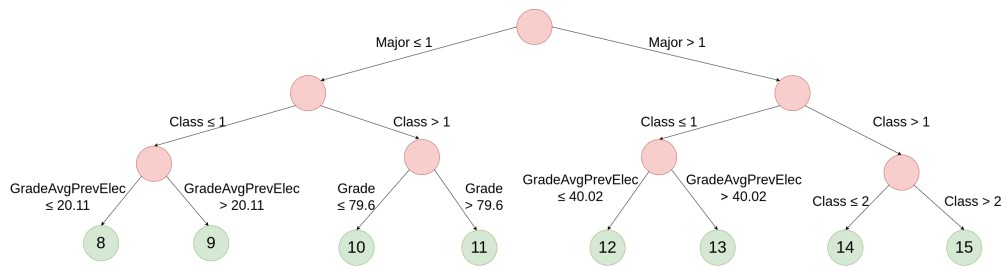

Figure 6: Visualization of an OCDT model on a *class* dataset.

text specifying the split feature and threshold. It is easy to observe on the tree plot that the OCDT made splits are both accurate and feasible. As we outline in Section C of the supplementary file, the target values are generated in a way that they are dependent on `Grade` and `GradeAvgPrevElec`. This is clearly depicted in the tree plot as the majority of splits employed by the OCDT do use those features. Additionally, the second-level splits in the tree plot are dependent on feature `Class` being one or higher, which aligns with the conditions imposing constraints to assign some targets to zero. This visualization example shows that OCDTs have the potential to be interpretable like their conventional counterparts available in standard DT implementations.

Our computational study shows that the proposed approach is highly effective in meeting constraints while delivering strong performance on various datasets. By incorporating optimization in split evaluations and leaf predictions, our methodology offers more accurate and feasible predictions compared to traditional methods, even on complex datasets. These results emphasize the usability and versatility of our approach, particularly in situations where compliance with constraints is essential for success.

## 4 CONCLUSION

We have introduced new methods to handle constraints among target variables in decision trees, an area yet unexplored in the existing literature. We have proposed an optimization-based exact approach and several heuristic approaches to ensure feasible predictions and maintaining computational efficiency. Through a computational study with existing and generated datasets, we have demonstrated the effectiveness of the proposed methods in producing feasible predictions, showcasing their potential for applications in multi-target learning.

**Limitations.** The work presented here has a few limitations. First, the optimization-based exact method can be computationally demanding for deep trees, which may impede its application for larger datasets. Second, the heuristic approaches proposed may not always produce accurate or even feasible predictions, limiting their reliability. Third, when the constraints among targets are highly non-convex, solving the prediction problem at each node can prove challenging, potentially affecting the overall performance of the decision tree. Finally, in its current form, it is unclear how these output-constrained decision trees can serve as the base learners for ensemble methods.

**Future research.** We aim to investigate the integration of our approach into bagging methods like random forests, as trees often act as base learners in these methods. We are also interested in exploring a warm-start method in the exact approach, where an optimization problem needs solving at each node. Given that these nodes share a parent-child relationship, the subsets they create could aid in devising this method for subsequent problems at the child nodes once the parent node is resolved. We are also considering the inclusion of target constraints in optimization-based algorithms for training trees like, for example, optimal classification trees (Bertsimas & Dunn, 2017). Although these algorithms can currently handle smaller problem sizes compared to standard implementations like CART (Breiman et al., 1984), this is a promising research direction. We are also looking into the potential advantages of considering constraints in clustering, since training a decision tree can be seen as a supervised clustering method.

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
