## Supplementary Material for "Output-Constrained Decision Trees"

## A   PROOF OF THEOREM 1

THEOREM 1 *Let the feasible set $\mathbb{Y} \subseteq \mathbb{R}^k$ be convex. If the loss function is the mean squared error or the mean Poisson deviance, then the mean of the target vectors in a node minimizes the loss.*

PROOF.   Suppose that the node corresponds to the subset $\mathcal{N} \subseteq \mathcal{D}$. Since $\mathbb{Y} \subseteq \mathbb{R}^k$, we have

$$\min_{\boldsymbol{y} \in \mathbb{R}^k} \sum_{i \in \mathcal{I}_\mathcal{N}} \ell(\boldsymbol{y}, \boldsymbol{y}_i) \leq \min_{\boldsymbol{y} \in \mathbb{Y}} \sum_{i \in \mathcal{I}_\mathcal{N}} \ell(\boldsymbol{y}, \boldsymbol{y}_i)$$

for any loss function $\ell : \mathbb{R}^k \times \mathbb{R}^k \mapsto \mathbb{R}$. We call the optimization model on the left as *the unconstrained problem* and the one on the right as *the constrained problem*. Note that if the optimal solution of the unconstrained problem is in the feasible set $\mathbb{Y}$, then it should also be the optimal solution of the constrained problem. We use the optimal solution of the unconstrained problem as the output vector of the node and refer to it as $\boldsymbol{y}_\mathcal{N}$. Both loss functions that we consider are separable in terms of the target vector. Thus, we write $\ell(\boldsymbol{y}, \boldsymbol{y}_i) = \sum_{t \in \mathcal{T}} \ell(y_t, y_{it})$, where $\mathcal{T} = \{1, \ldots, k\}$ is the set of target indices.

The unconstrained problems with the mean squared error and the mean Poisson deviance loss functions are given by

$$\min_{\boldsymbol{y} \in \mathbb{R}^k} \frac{1}{2|\mathcal{I}_\mathcal{N}|} \sum_{i \in \mathcal{I}_\mathcal{N}} \sum_{t \in \mathcal{T}} (y_t - y_{it})^2$$

and

$$\min_{\boldsymbol{y} \in \mathbb{R}^k} \frac{1}{|\mathcal{I}_\mathcal{N}|} \sum_{i \in \mathcal{I}_\mathcal{N}} \sum_{t \in \mathcal{T}} (y_{it} \ln(\frac{y_{it}}{y_t}) + y_t - y_{it}),$$

respectively. Clearly, both loss functions are differentiable and convex. Thus, the necessary-and-sufficient optimality conditions for both problems are given by

$$\sum_{i \in \mathcal{I}_\mathcal{N}} \frac{\partial \ell(y_t, y_{it})}{\partial y_t} = 0, \;\; t \in \mathcal{T}. \tag{7}$$

Solving next the conditions in (7) for both unconstrained problems leads to

$$\sum_{i \in \mathcal{I}_\mathcal{N}} \frac{\partial \ell(y_t, y_{it})}{\partial y_t} = 0, \; t \in \mathcal{T} \implies y_t = \frac{1}{|\mathcal{I}_\mathcal{N}|} \sum_{i \in \mathcal{I}_\mathcal{N}} y_{it}, \; t \in \mathcal{T} \implies \boldsymbol{y}_\mathcal{N} = \frac{1}{|\mathcal{I}_\mathcal{N}|} \sum_{i \in \mathcal{I}_\mathcal{N}} \boldsymbol{y}_i.$$

Thus, the optimal solutions is the mean vector. Because $\boldsymbol{y}_i \in \mathbb{Y}$ for $i \in \mathcal{I}_\mathcal{N}$ and the feasible set $\mathbb{Y}$ is convex, the convex combination of $\boldsymbol{y}_i$ vectors should also be feasible; that is, $\boldsymbol{y}_\mathcal{N} \in \mathbb{Y}$. Therefore, the mean vector is also the optimal solution of the constrained problem. This shows the desired result. ∎

**Remark 1** *Standard multi-target regression tree implementations, like that of* `scikit-learn`, *use the median of the target vectors in the node as prediction when the loss function is selected as the mean absolute deviation. However, our result in Theorem 1 does not hold in this case, because the median vector may lie outside the feasible set even if it is convex.*

*Figure 7 shows such an example for $k = 2$, where the* convex *feasible set is shown by the shaded area, and the node has five feasible target vectors denoted by the black circles. The median vector, shown as the red circle, is clearly outside the feasible set.*

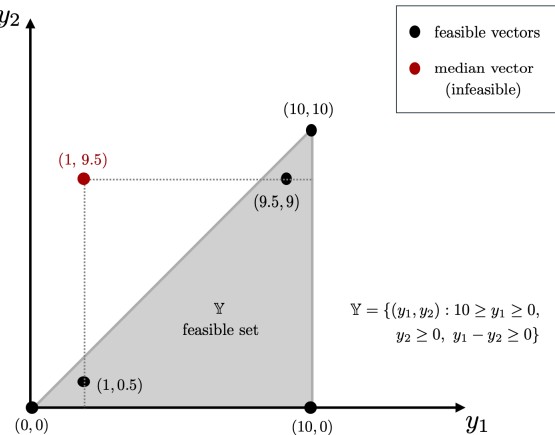

Figure 7: A counter example showing that the median vector lies outside the *convex* feasible set.

## B    OPTIMIZATION MODEL OF THE *scores* DATASET

The optimization model imposed on the *scores* dataset is formulated as follows:

$$
\begin{aligned}
\text{minimize} \quad & \frac{1}{2n} \sum_{i=1}^{n} \sum_{t=1}^{k} (y_t - y_{it})^2 \\
\text{subject to} \quad & y_t \leq 110 b_t, \qquad t = 1, \dots, k, \\
& y_2 \geq 50 b_3, \\
& y_1 + y_2 \geq 110 b_1,
\end{aligned}
$$

where $n$ is the number of instances, $k$ is the number of targets, $y_t$ is the prediction for the $t$-th target, $y_{it}$ is the true value of the $t$-th target for the $i$-th instance, and $b_t$ is a binary variable indicating whether the prediction for the $t$-th target is used ($b_t = 1$) or not ($b_t = 0$).

## C    DATA GENERATION AND OPTIMIZATION MODEL OF THE *class* DATASET

Table 1 outlines the distributions and their parameters used to generate the extended data, illustrating how the features of the dataset are created. We should note that the feature `GradePerm` consists of the permuted values of the feature `Grade`, created with the purpose of imposing randomness on the data. The target values are computed by taking the equally weighted sum of the `Grade` and `GradeAvgPrevElec`, with the addition of a uniform noise component ranging between -10 and 10 if `GradeAvgPrevElec` is non-zero. If `GradeAvgPrevElec` is zero, the target value is determined by `Grade` plus a normal noise with a mean of 10.0 and a standard deviation of 2.0. These computed values are then clipped to ensure they fall within the range of 0 to 100. Additionally, certain target values are deliberately set to zero based on random selection according to the features. For instance, if the `Class` feature is equal to one, all but a randomly chosen subset of the first two targets' grades are set to zero. To fit the data to our research objectives, we have imposed two constraints on the created dataset. One constraint is created using binary decision variables to ensure that one non-zero prediction is generated at most. Another set of constraints has been applied to each target variable to limit the prediction based on the binary variable.

Table 1: Features of the *class* dataset

| Features | Distribution | Min | Max | Mean | Std.Dev. |
|---|---|---|---|---|---|
| EnrolledElectiveBefore | Discrete Uniform | 0 | 1 | - | - |
| GradeAvgPrevElec | Normal | 0 | 100 | 60 | 15 |
| Grade | Normal | 0 | 100 | 70 | 10 |
| Major | Discrete Uniform | 1 | 3 | - | - |
| Class | Discrete Uniform | 1 | 4 | - | - |
| GradePerm | Normal | 0 | 100 | 70 | 10 |

The optimization model is as follows:

$$\text{minimize} \quad \frac{1}{2n} \sum_{i=1}^{n} \sum_{t=1}^{k} (y_t - y_{it})^2$$

$$\text{subject to} \quad \sum_{j=1}^{k} b_j \leq 1,$$

$$y_t \leq 110 b_t,$$

where $n$ is the number of instances, $k$ is the number of targets, $y_t$ is the prediction for the $t$-th target, $y_{it}$ is the true value of the $t$-th target for the $i$-th instance, and $b_t$ is a binary variable indicating whether the prediction for the $t$-th target is used ($b_t = 1$) or not ($b_t = 0$).