# OpenReview forum: "Output-Constrained Decision Trees"
_ICLR.cc/2025/Conference — Submitted to ICLR 2025_

### Official Review · Reviewer_qMoe · 2024-11-03

**Soundness:** 3
**Presentation:** 2
**Contribution:** 2
**Rating:** 5
**Confidence:** 3

**Summary:**

As far as I understand, this submission proposes a framework for multivariate regression trees under output constraints. Under this setting, the representative output vector of each node or leaf should belong to some given feasible set, and should be as accurate as possible with respect to the evaluation loss. I think this setting is interesting and can be useful in different applications, provided that the constraints, that shape the feasible sets, are well-defined.

The submission elaborates on the problem of finding a representative output vector both exactly and approximately/heuristically. Experiments on two data sets are conducted, where the loss is the mean squared error (MSE), to assess the potential advantages of the proposed framework.

**Strengths:**

S1: I think learning and predicting under output constraints are interesting problems.

S2: The problem of finding a representative output vector, which is an essential step of the proposed framework, is elaborated both exactly and approximately/heuristically.

S3: Experiments are conducted to assess the potential advantages of the proposed framework.

**Weaknesses:**

W1: The problem statement is not entirely clear. In particular, it might be beneficial to further clarify whether the constraints are imposed at the instance level, i.e., different instances can have different sets of feasible outputs, or at the population level, i.e., all the instances should have the same set of feasible outputs. I think neither of the cases calls for further elaboration/clarification on the prediction phase. However, the earlier case might call for further elaboration/clarification on the training phase.

W2: I think more insight discussions about the potential (dis)advantages of the proposed framework might be beneficial. The author(s) might consider adding the (average) scores on the cases DTs provide feasible and infeasible solutions separately. Ideally, one might wish to see frameworks/methods, which provide promising scores on both parts, but a good trade-off between the two parts might also be appreciated.

W3: More discussions on extensions to ensemble methods and adaptations to multi-label and multi-dimensional classification problems might be beneficial. For example, while the submission mentions the label powerset algorithm as a solution for multi-label and multi-dimensional classification problems, handling a possibly large multi-class classification problem, which can be both data-hungry and contain output constraints at the instance level might not be obvious.

**Questions:**

Q/S1: Do you assume the constraints are imposed at the instance level, i.e., different instances can have different sets of feasible outputs, or at the population level, i.e., all the instances should have the same set of feasible outputs? Please refer to W1 for my further comments on this point.

Q/S2: It might be beneficial to show the (average) scores on the cases where the DTs provide feasible and infeasible solutions separately.

Q/S3: While extensions to ensemble methods and adaptations to multi-label and multi-dimensional classification problems seem to be not obvious, adding more discussions on potential directions would be appreciated.

---

### Official Review · Reviewer_8Ds6 · 2024-11-03

**Soundness:** 2
**Presentation:** 2
**Contribution:** 2
**Rating:** 3
**Confidence:** 4

**Summary:**

The paper considers the problem of learning decision trees where the leaf nodes predict (constant) vector-valued output with some constraint on them, for example $\ell_0$ constraint. It adapts the traditional CART-type greedy top-down induction method and when evaluating each possible split (feature index and threshold pair), it takes into account the constraints. The split evaluation is either solved exaclty (e.g. using mixed-integer optimization solvers), or approximately using a penalty method or using a best training point as a solution. The proposed method has been evaluated on smaller-scale problems.

**Strengths:**

The proposed heuristic approaches are interesting though not thoroughly explored or analyzed.

**Weaknesses:**

1. Unprecise problem formulation. The paper discusses constraints on leaf predictions in general without specifying the exact type of constraints considered. Some constraints are much harder (or easier) to solve than others.
2. Limited algorithmic novelty. The proposed method is still based on the traditional greedy top-down induction trees, and just adapts the split evaluation to take into account the constraints.
3. Lack of comparison. Methods that attempt to learn exactly (optimally) decision trees can straightforwardly incorporate the constraints in the objective function (and perhaps this makes the search space smaller and accelerates those type of algorithms). Given that the paper considers smaller-scale problems (1000s training points), those optimal decision tree algorithms can be trained for these problems, and thus be compared.
4. The literature review does not consider methods on fairness where different type of constraints are used for learning decision trees. It also does not consider other methods for learning trees such as dynamic programming, MIO and alternating optimization.

**Questions:**

No questions.

---

### Official Review · Reviewer_eDCm · 2024-11-04

**Soundness:** 2
**Presentation:** 2
**Contribution:** 2
**Rating:** 5
**Confidence:** 5

**Summary:**

The authors propose an algorithm for regression problems with constraints on tree outputs. Their approach modifies the greedy recursive partitioning algorithm to grow a binary tree until a specified maximum depth or minimum number of samples in a node is reached. To split a node, the algorithm uses a mixed integer optimization (MIO) solver to find a feasible vector that minimizes the mean squared error loss. To reduce computation time, the authors introduce two heuristics: (1) Lagrangian relaxation and (2) selecting a feasible medoid as an approximation. The experiments are conducted on a single dataset, modified to evaluate the algorithm under various regimes (e.g., different number of constraints and dataset sizes). The method is compared against variations of itself, differing in the types of node optimization (e.g., with or without relaxation), and a baseline CART-like tree where outputs are replaced with feasible values.

**Strengths:**

1. The proposed algorithm addresses an interesting and practical problem of a regression with constraints on outputs.
2. The authors propose heuristics to alleviate the computational time of MIO.

**Weaknesses:**

1. The use of an MIO solver within the tree induction process raises concerns about practicality in real-world applications, as the MIO problem is NP-hard and may not terminate in a reasonable amount of time.
2. The training times are reported in scaled units, which makes it difficult to see the true computational cost of the algorithm.
3. The paper uses a single dataset, making it difficult to assess performance relative to a straightforward baseline (e.g., training a CART regression tree and substituting outputs with feasible values).

**Questions:**

N/A

---

### Official Review · Reviewer_ouj9 · 2024-11-04

**Soundness:** 2
**Presentation:** 1
**Contribution:** 2
**Rating:** 3
**Confidence:** 3

**Summary:**

This paper investigates multi-target learning problems where constraints are imposed on the target variables. To address this challenge, the authors propose an exact optimization-based approach that enforces these constraints during the decision tree-building process, ensuring feasibility but at the cost of increased computational complexity. Additionally, they introduce two heuristic methods that offer more efficient alternatives. Through experiments on datasets with imposed constraints, the authors demonstrate the effectiveness of these approaches in maintaining feasible predictions under constrained settings.

**Strengths:**

1. The paper poses an insightful and valuable research question regarding constrained multi-target learning in decision trees and presents effective approaches to address it.
2. The paper explain the problem and the methods clearly and intuitively.

**Weaknesses:**

1. The paper would benefit from a more robust differentiation of its contributions from existing research, particularly through a deeper exploration of recent work. The Related Work section includes only a few papers from the last five years, which limits the context for evaluating the novelty of this approach. Additionally, the claim that this is the first method to address target-constrained problems may be overstated. Expanding on prior work in this area would help to clarify the unique aspects and positioning of this contribution.
2. In the SCORE dataset, two specific constraints were imposed on the target variables, but the rationale behind selecting these particular constraints is unclear. Could the authors provide justification for the appropriateness of these constraints for this dataset? If the constraints are too tailored to the proposed method, this may obscure potential limitations.
3. It would be clear if the paper could provide intuition behind the two heuristic methods and explicitly highlight the benefit of each heuristic from a design perspective.
4. The reliance on a self-modified dataset and a baseline comparison with only the traditional decision tree limits the strength of the paper’s claims regarding novelty and contributions. Have the authors considered testing the proposed methods against state-of-the-art decision tree models or advanced tree-based methods that incorporate constraints? Broader comparisons with such models would provide a stronger foundation for evaluating the effectiveness and competitive advantage of the proposed approaches.
5. The paper lacks a detailed analysis of the computational advantages of the two heuristics. It would be helpful to specify the order of magnitude of the improvement and to quantify how many seconds the heuristics take relative to a traditional decision tree. Providing these details would clarify the efficiency gains of the heuristics.

**Questions:**

Besides comments in Weaknesses section, I also list several questions here.
1. The experiments are based on artificial constraints applied to a single dataset. Could the authors suggest specific real-world datasets where such constraints naturally occur? Additionally, could the authors discuss how their approaches might be adapted or evaluated in those contexts?
2. Since the dataset was modified to include specific constraints, it is unclear how the insights derived from Figure 1, particularly regarding the trade-off, remain reasonable and reliable. Adjusting a dataset with imposed constraints can introduce biases or alter its original distribution, potentially influencing the model’s behaviours. It would be beneficial for the authors to discuss methods for mitigating these potential biases to ensure the robustness of their findings or provide a comparative analysis between the original and modified datasets, showing how the distribution of key variables changed.
3. Could you clarify which specific experiment led to the results in Figure 1? In particular, it is not immediately clear how the 4000 samples were generated. It would be better to provide a specific section reference where the experiment details should be added or include a brief description of the experimental setup directly in the figure caption.

---

### Meta-Review · Area_Chair_vbPc · 2024-12-21

**Metareview:**

The paper concerns the problem of multi-target learning in which constraints are imposed on target variables. The Authors introduce a variant of the decision tree algorithm to solve this problem.

The Reviewers highlighted several weaknesses in the submission, including an imprecise problem definition, a limited discussion of related work, insufficient evidence regarding computational costs, and a lack of extensive empirical studies.

All ratings are below the bar and the Authors did not prepare any rebuttal.

**Additional Comments On Reviewer Discussion:**

There was no discussion as the Authors did not sent rebuttal.

---

### Decision · Program_Chairs · 2025-01-22

Reject